# Oral Administration of Okara Soybean By-Product Attenuates Cognitive Impairment in a Mouse Model of Accelerated Aging

**DOI:** 10.3390/nu11122939

**Published:** 2019-12-03

**Authors:** Henry M. Corpuz, Misa Arimura, Supatta Chawalitpong, Keiko Miyazaki, Makoto Sawaguchi, Soichiro Nakamura, Shigeru Katayama

**Affiliations:** 1Interdisciplinary Graduate School of Science and Technology, Shinshu University, 8304 Minamiminowa, Kamiina, Nagano 3994598, Japan; 15st553b@shinshu-u.ac.jp (H.M.C.); snakamu@shinshu-u.ac.jp (S.N.); 2Rice Chemistry and Food Science Division, Philippine Rice Research Institute, Maligaya, Science City of Muñoz, Nueva Ecija 3119, Philippines; 3Faculty of Agriculture, Shinshu University, 8304 Minamiminowa, Kamiina, Nagano 3994598, Japan; 17as202g@shinshu-u.ac.jp (M.A.); supattra.chawalitpong@gmail.com (S.C.); 4Misuzu Corporation Co., Ltd., 1606 Wakasato, Nagano City, Nagano 3800928, Japan; nakazawa-k@misuzu-co.co.jp (K.M.); sawamako@misuzu-co.co.jp (M.S.); 5Interdisciplinary Cluster for Cutting Edge Research, Shinshu University, 8304 Minamiminowa, Kamiina, Nagano 3994598, Japan

**Keywords:** okara, cognitive impairment, neuroprotection, BDNF, SAMP8

## Abstract

The microbiota–gut–brain axis has attracted increasing attention in the last decade. Here, we investigated whether okara, a soybean by-product rich in dietary fiber, can attenuate cognitive impairment in senescence-accelerated mouse prone 8 (SAMP8) mice by altering gut microbial composition. Mice were fed either a standard diet, or a diet containing okara (7.5% or 15%, *w*/*w*) for 26 weeks. In the memory test, the 7.5% okara-fed mice showed a longer step-through latency and the 15% okara-fed mice had a short escape latency compared with control mice. The 15% okara-fed mice displayed decreased body weight, increased fecal weight, and altered cecal microbiota composition compared with the control group; however, there was no significant difference in the serum lactic acid and butyric acid levels among these mice groups. The 7.5% okara-fed mice had significantly higher NeuN intensity in the hippocampus compared with control mice. Furthermore, a decrease in inflammatory cytokine TNF-α and an increase in brain-derived neurotrophic factor (BDNF) was observed in the 7.5% okara-fed group. The expression of synthesizing enzyme of acetylcholine was increased by the okara diets, and the acetylcholine level in the brain was higher in the 7.5% okara-fed group than in the control. These suggest that oral administration of okara could delay cognitive decline without drastically changing gut microbiota.

## 1. Introduction

Microbiota–gut–brain axis is a bidirectional communication system between the brain and gastrointestinal tract, linking emotional and cognitive centers of the brain with peripheral intestinal functions [1]. Since the gut microbiota plays an important role in host metabolism and intestinal inflammation responses, dysregulation of microbiota and their excessive endotoxin production cause endotoxemia and systemic inflammation, resulting in neuropsychiatric disorders [2,3]. Disruption in the gut–brain axis is also known to influence mood and cognitive performance [4,5].

Dietary fiber, a prebiotic source, is a key component in healthy eating and plays an important role in physiological processes as well as disease prevention. Prebiotics are defined as dietary or complex carbohydrates, which escape digestion in the small intestine and are fermented by bacteria in the large intestine. This beneficially enhances the growth of exogenous or endogenous probiotics or commensal microbes [6,7]. Probiotics are live microorganisms that confer a health benefit to the host when administered in adequate amounts [8]. Prebiotic-induced enhancement of these microbes has many beneficial effects on host immunity and physiology [9]. The combined or synergistic effects of prebiotics and probiotics are known as synbiotics. Mounting evidence suggests that the bidirectional interaction between gut microbiota and the brain can be modulated by probiotics, prebiotics, synbiotics, and diet, which exert positive effects on brain activity, behavior, and cognition [10]. High fiber diets have been associated with several health benefits, including a reduction in the risk of type 2 diabetes, colon cancer, obesity, stroke, and cardiovascular diseases [11]. Aside from the benefits of dietary fiber in the prevention of chronic diseases, it also contributes to the improvement of cognitive function and brain health. Additionally, recent studies have demonstrated that prebiotics can modulate brain function and signaling activity [10,12]. For example, oral administration of the synbiotic combination of probiotic *Enterococcus faecium* and prebiotic inulin increased short-chain fatty acid butyrate production and subsequently provoked an increase in brain-derived neurotrophic factor (BDNF) and a decrease in pro-inflammatory cytokine concentrations in the hippocampus [13]. Intake of isolichenan, an α-glucan from the lichen (*Cetrariella islandica*) reversed ethanol-induced memory impairment in mice [14]. Supplementation of arabinoxylan from the yeast *Triticum aestivum* and β-glucan from barley were able to preserve memory in a mouse model of vascular dementia [15].

The senescence-accelerated mouse (SAM) is a murine model for accelerated aging [16]. Among the strains, SAM prone 8 (SAMP8) is characterized as a mouse model for age-related cognitive impairment [17,18]. Many studies have demonstrated that the oral intake of probiotic strains and food bioactive compounds can suppress the cognitive impairments in SAMP8 mice [19,20,21]. Okara is a by-product generated from soymilk or tofu production in large quantities and is mainly composed of insoluble dietary fiber (approximately 50%) in the form of cellulose and hemicellulose [22]. Okara is utilized in East Asian countries as a food ingredient due to its high fiber content. To date, the preventive effect of okara against metabolic disorders such as diabetes, obesity, and hyperlipidemia has been reported [23]. However, its potential action in the prevention of age-related cognitive decline is poorly understood. In the present study, we evaluated the effects of the oral administration of okara on age-related cognitive decline in SAMP8 mice. The effect of okara administration on gut microbiota was also investigated to determine whether dietary fiber-rich okara can affect cognitive performance by altering the gut microbiota.

## 2. Materials and Methods

### 2.1. Animals

Male SAMP8 and senescence-accelerated resistant mouse 1 (SAMR1) mice at 16 weeks of age were purchased from Japan SLC, Inc. (Shizuoka, Japan). Mice were housed individually and maintained in a temperature-controlled (20−23 °C) and a humidity-controlled (40–70%) animal room with an alternating 12 h/12 h light-dark cycle (lights on between 7:00 and 19:00). All animal experiments and protocols were approved by the Institutional Animal Care and Use Committee of Shinshu University (Permission No. 270076).

### 2.2. Animal Protocol

SAMP8 mice were divided into 3 groups: control, okara 7.5%, and okara 15% (*n* = 11–15). SAMR1 mice were used as the normal aging group (*n* = 8). The SAMP8 control group and SAMR1 group were fed an AIN-93M diet (Oriental Yeast, Tokyo, Japan), and the mice in the last two groups were fed an AIN-93M diet containing 7.5% and 15% okara (*w*/*w*). Okara was provided from Misuzu Corporation Co., Ltd. (Nagano, Japan). The composition of okara was as follows: 23.7% protein, 47.8% carbohydrate, 18.7% lipid, 44.5% dietary fiber, and 0.072% isoflavone. To normalize the nutritional value of the diet, the amounts of constituents were adjusted based on the nutritional compositions of okara (Table 1). The mice were allowed free access to food and tap water. Body weight and food intake were recorded every week and every two weeks, respectively. After 26 weeks of feeding, the cognitive performance of the mice was assessed using Barnes maze and passive avoidance tests. After the last test, the feces were collected and the mice were sacrificed. The blood, hippocampus, and cecum were collected, frozen in liquid nitrogen, and kept at −80 °C before analysis.

### 2.3. Barnes Maze Test

The spatial learning and memory ability of mice was evaluated using the Barnes maze test as previously described [19]. The escape latency to reach the escape box and staying time in each quadrant were recorded during the probe test. Good memory performance is reflected by shorter escape latency and stay time at the target quadrants where the escape box is located.

### 2.4. Passive-Avoidance Task

The acquisition and memory retention of the mice were determined using the passive avoidance task. Testing was done using a step-through test cage (Muromachi Kikai Co. Ltd., Tokyo, Japan) consisting of white and black compartments separated by a sliding door. In the acquisition trial, the mice were initially placed in the white compartment. The door between the two compartments was opened 10 s later and was closed when the mice entered the black compartment. A 3 s electrical foot shock (0.3 mA) was then delivered through the stainless-steel rods. Probe test was conducted 24 h after the acquisition trial by returning the individual mice to the illuminated compartment. Step-through latencies were recorded for up to 300 s during the probe test. Step-through latency was defined as the time it took for the mouse to enter the dark compartment after the door was opened.

### 2.5. Microbiota Sequencing and Analysis

Cecal microbial analysis was conducted using the terminal restriction fragment length polymorphism (T-RFLP) method by the Central Institute for Experimental Animals (ICLAS: Kawasaki, Kanagawa, Japan). The fecal microbiota test animals were analyzed using tag-encoded 16S rRNA gene high-throughput sequencing. Briefly, DNA isolated from the caecum was used for T-RFLP and was subjected to 16S rDNA. ABI PRISM 310 Genetic Analyzer (Applied Biosystems, Carlsbad, CA, USA) and GeneScan (Applied Biosystems) were used for fragment analysis. The length of each fragment was determined by the operational taxonomic unit (OTU) and estimated primary microbiota was based on the ICLAS database for mice.

### 2.6. Measurement of Lactic Acid and Butyric Acid Concentration

The concentration of lactic acid and butyric acid in the serum was measured by HPLC. Briefly, standard lactic acid and butyric acid and the serum samples, including 4-methylvaleric acid as an internal standard, were labeled with an SCFA labeling kit (YMC Co., Ltd., Kyoto, Japan), separated on a YMC-Pack FA column (250 × 6.0 mm ID; YMC Co. Ltd.,) and detected by UV absorbance at 400 nm.

### 2.7. Quantitative Real-Time PCR (qRT-PCR) Analysis

Hippocampal gene expressions were measured by qRT-PCR assay. Total RNA was extracted using TRizol reagent kit (Invitrogen, Carlsbad, CA, USA), and cDNA was synthesized using ReverTra Ace qPCR RT Master Mix with gDNA Remover Kit (Toyobo, Osaka, Japan) according to the manufacturer’s instructions. qRT-PCR was performed using a Kapa SYBR Fast qPCR kit (Kapa Biosystems, Woburn, MA, USA) on a Thermal Cycler Dice Real-Time System (TaKaRa Bio, Shiga, Japan). The housekeeping gene β-actin was used as an internal control, and the relative expression was expressed as the fold change compared to the control samples using the comparative 2^−ΔΔ*C*t^ method. The primer sequences used for RT-qPCR are listed in Appendix A.

### 2.8. Western Blotting Analysis

The expression of BDNF protein in the hippocampus was evaluated by Western blot analysis. The concentrations of proteins extracted with TRizol reagent (Thermo Fisher Scientific, Inc., Tokyo, Japan) were determined using Bradford’s method with bovine serum albumin (BSA) as a standard. Each amount of proteins was subjected to SDS-PAGE and then transferred to polyvinylidene difluoride membranes (0.45 µm, Merck Millipore, Billerica, MA, USA). The membranes were blocked with 3% BSA in Tris-buffered saline (TBS) with 0.05% Tween-20 (TBST) for 1 h at room temperature, and incubated overnight at 4 °C with primary antibodies against BDNF (1:3000; Abcam, Cambridge, MA, USA) and tubulin (1:5000; Abcam). Next, the membrane was washed with TBST three times and incubated with secondary horseradish peroxidase-conjugated antibody (1:5000; Santa Cruz Biotechnology, Santa Cruz, CA, USA) at room temperature for 1 h. The protein bands were visualized using the EzWestLumi plus kit (ATTO, Tokyo, Japan) and AE-9300 Ez-Capture (ATTO). The intensity of protein bands was analyzed using ImageJ software (NIH Image, Bethesda, MD, USA).

### 2.9. Immunohistochemical Analysis

For NeuN staining, paraffin-embedded mouse brain sections were dewaxed using xylene and hydrated in ethanol at decreasing concentrations. The sections were boiled in 10 mM Tris/1 mM EDTA buffer (pH 9.0) for 20 min and cooled down for 30 min at room temperature for antigen retrieval. The sections were washed two times with TBST solution. After blocking with 5% BSA in TBST for 1 h, the sections were incubated with antibodies against NeuN (1:200, Abcam) overnight at 4 °C. The slides were washed two times in TBST and incubated with the secondary antibody, Alexa Fluor 488 goat anti-rabbit IgG (H&L) (1:100, Abcam). After washing two times with TBST solution, the sections were mounted with immunoselect antifading mounting medium DAPI (Dianova, Hamburg, Germany) and the NeuN-immunoreactive cells were examined under a fluorescence microscope (EVOS FL; Advanced Microscopy Group, Bothell, WA, USA).

For choline acetyltransferase staining, the brain tissue sections were deparaffinized by using clear plus solution (Falma, Tokyo, Japan) for three times (3 min/time) and rehydrated with 100%, 95%, 75%, 50% ethanol, and DI water, respectively. Tris/EDTA buffer pH 9 was used for antigen activation at 100 °C for 10 min. The processed tissue samples were blocked with 10% FBS serum and 1% BSA in TBS for 1 h, probed with anti-choline acetyltransferase (1:1000, Abcam) for overnight and washed with 0.05% Tween20 and 1% BSA in TBS for two times (5 min/time). Alexa Fluor 555 rabbit anti-sheep IgG (Fab2) (1:200, Abcam) and DAPI were stained. The expression of choline acetyltransferase was detected using the EVOS FL Auto Imaging System (Thermo Fisher Scientific).

### 2.10. Measurement of Acetylcholine Concentration

The concentration of acetylcholine in the brain was measured using a commercial Amplite^TM^ Fluorimetric Acetylcholine Assay Kit (AAT Bioquest Inc., Sunnyvale, CA, USA) according to the manufacturer’s instructions.

### 2.11. Statistical Analysis

The GraphPad Prism 5.0 software (La Jolla, CA, USA) was used to perform statistical analyses. Data are represented as the means ± standard error of the mean (SEM). Differences between the means were evaluated using ANOVA followed by the Bonferroni or Tukey’s post hoc test for mean comparisons.

## 3. Results

### 3.1. Effects of 26 Weeks of Okara Administration on the Body Weight, Food Intake, and Fecal Weight of SAMP8 Mice

Consumption of foods rich in dietary fiber is associated with reduced energy intake, body weight gain, and improvement of gut commensal microbiota. To determine the effect of okara on body weight management, SAMP8 mice were fed with either a normal diet (AIN-93M) or a diet supplemented with okara for 26 weeks. Significant reduction in the body weight of mice fed a diet supplemented with 15% okara was observed from the third week of treatment without any changes in feed consumption throughout the experiment (Figure 1A,B). The weight of collected fecal matter of 42-week-old mice was also significantly higher for okara-fed mice compared to the control group, and depended on the dosage (Figure 1C). These results suggest that administration of higher doses of okara might be effective in decreasing body weight and increasing the number of feces.

### 3.2. Oral Administration of Okara Suppressed Age-Related Cognitive Impairment in SAMP8 Mice

The effect of long-term administration of a diet supplemented with okara on the cognitive performance of SAMP8 mice was evaluated using Barnes maze and passive avoidance tests, which are dependent on hippocampal function. After 26 weeks of receiving a diet supplemented with 15% okara, escape latencies were significantly lower compared to the control group during the training sessions and probe test (Figure 2A,B). In addition, the time spent in the target quadrant during the probe test in the okara treated mice was also higher; however, it was not significantly different from the control group (Figure 2C). On the contrary, the amount of time spent in the opposite of target was significantly lower in the 7.5% and 15% okara groups compared with the control group. A fear-motivated passive-avoidance test was also performed to examine the learning and short-term memory abilities of the SAMP8 mice. The step-through latency of the mice fed a diet supplemented with 7.5% okara was significantly higher than those of the control group (Figure 2D). Taken together, these results suggested that long-term continuous intake of okara could prevent age-related cognitive decline in SAMP8 mice.

### 3.3. Administration of Okara at a Higher Dosage Altered the Cecal Microbial Composition of SAMP8 Mice

We next investigated the bacterial composition and microbial diversity of the cecum in mice fed with the differing diets for 26 weeks. Hierarchical clustering showed that the microbial composition of the cecum in the okara-treated mice was distinguishable from that of the control groups (Figure 3A). Specifically, the okara fed mice showed a distinct microbial composition that clustered separately from that of the control group and the SAMR1 control strain mice (Figure 3B). The relative abundance of the microbial composition of mice fed with the high dose okara diet was the most different from the control mice. Figure 3C shows the most abundant bacterial communities at the phylum level. The relative abundance of *Clostridiales*, *Bacteriodales*, and *Ciriobacteriales* were significantly increased by 15% okara diet supplementation compared to control mice. In contrast, *Lactobacillus*, *Erysipelotrichaceae*, *Parasutterella*, and other unclassified bacterial strains were significantly decreased in the 15% okara group compared to the control group (Figure 3C). The content of lactic acid and butyric acid in the serum of mice was measured using HPLC. As shown in Figure 4A, the lactic acid content was lower in the 15% okara group than in the control group, but this decrease was not statistically significant. On the other hand, there were no significant differences in butyric acid content between control and okara groups (Figure 4B).

### 3.4. Neuroprotection in the Hippocampus and Dentate Gyrus (DG) of SAMP8 Mice by Administration of a Diet Supplemented with Okara

Immunostaining of brain sections showed that the NeuN intensity of both the hippocampus and DG of SAMP8 control nice was lower than that of SAMR1, suggesting that there was significant nerve damage in the brain (Figure 5). On the other hand, the NeuN intensity of both the hippocampus and DG was significantly higher in mice treated with the 7.5% okara-supplemented diet. However, the NeuN intensity remained unchanged between the control and 15% okara groups. We then measured the expression levels of neuroinflammation and neuroprotection-associated factors in the hippocampus after the okara diet had been administered. For neuroinflammatory factors, the expression level of TNF-α was decreased in the okara-fed groups compared with the control group (Figure 6A). For the neuroprotective factors, the gene expression levels of BDNF and NT-3 in the SAMP8 control group were significantly lower than those of the SAMR1 group; however, okara administration increased the gene expression of BDNF and NT-3 (Figure 6B). The gene expression level of immunosuppressive IL-10 was higher in the okara-fed groups than in the SAMP8 control group; however, these increases were not statistically different between the mice groups. A significant increase in BDNF protein expression was observed in the 7.5% okara group compared with the control group (Figure 6C).

### 3.5. Effects of Okara Administration on the Cholinergic System in the Mouse Hippocampus

It has been proposed that the dysfunction of the cholinergic system in the hippocampus and cortex contributes to the learning and memory deficits associated with aging and Alzheimer’s disease [24]. Hence, we evaluated the modulating effects of okara on key enzymes involved in acetylcholine synthesis. As shown in Figure 7A, okara administration increased the mRNA expression levels of choline acetyltransferase (ChAT), an acetylcholine-synthesizing enzyme, in the hippocampus of mice. In contrast, the mRNA expression level of acetylcholine esterase (AChE), an enzyme responsible for the degradation of acetylcholine, was not affected by okara diet treatment. Furthermore, okara-fed mice had higher ChAT staining in the hippocampus compared to the control group (Figure 7B). In the whole brain, higher acetylcholine level was observed in the SAMR1 and 7.5% okara-fed groups compared to the control group, whereas the 15% okara-fed group showed the same level as control group (Figure 7C).

## 4. Discussion

The present study demonstrated that dietary supplementation with the dietary fiber-rich okara soybean by-product for 26 weeks attenuated age-related cognitive impairment in SAMP8 mice. Prebiotic and probiotic intake has been suggested as a promising approach to improve human health by modifying and enhancing the composition of the gut microbiome, reducing intestinal inflammation and decreasing energy harvest [25]. Here, we showed that the administration of a higher dosage of okara (15%, *w*/*w* in a diet) resulted in decreased body weight, increased fecal weight, and alteration of the gut microbiota. We also observed an increase in the abundance of *Clostridiales* bacteria in the ceca of the 15% okara group. Several specific gut bacteria species play a role in host resistance against intestinal pathogens and promote systemic autoimmunity [25]. Short-chain fatty acids (SCFAs) are produced as metabolites of dietary fiber by intestinal bacteria. SCFAs have been reported to exhibit immunomodulatory effects and contribute to the maintenance of intestinal health. Clostridia produce butyrate, which induces the differentiation of regulatory T cells and contributes to the maintenance of gut immune homeostasis and the prevention of colitis and allergic responses [26,27,28]. Additionally, recent studies have shown a relationship between gut microbiota and brain function and that butyrate production by microorganisms activates the BDNF secretion [29,30]. However, there was no significant difference in the serum butyric acid level between control and 15% okara groups. Therefore, the alteration in the microbiota after administration of a high dose okara might not affect the maintenance of intestinal immune homeostasis and brain function.

Aging is associated with increased oxidative stress and inflammation leading to cognitive dysfunction. As gut permeability increases with age, the intestinal microbiota plays an important role in initiating these factors. Usually, leaky gut microbes and microbial contents escape and translocate into the bloodstream. This leads to activation of the immune system and the release of pro-inflammatory cytokines, as well as an increased load of circulatory pro-inflammatory markers. The continued exposure of the brain to these inflammatory factors could lead to cognitive decline [31]. Neuroinflammation is associated with cognitive impairment and memory decline, as the hippocampus is vulnerable to altered synaptic transmission and plasticity during inflammation [32]. On the other hand, neurotrophic factors such as BDNF and NT-3 play an important role in the health of neurons such as normal neural development, the survival of existing neurons, and activity-dependent neuroplasticity in the brain [33]. In this study, the expression of TNF-α in the hippocampus was attenuated by the administration of an okara diet, while the increased BDNF expression was observed in the okra-fed group. Thus, the possible neuroprotective effects of okara against cognitive dysfunction might be caused by the suppression of age-related neuroinflammation.

Cholinergic deficits, as well as neuroinflammation, are known to be involved in the progression of neurodegenerative diseases such as Alzheimer’s disease [34], suggesting that acetylcholine is vital in maintaining synaptic plasticity and mediating learning and memory [35]. In this study, we showed that there was an increase in the acetylcholine content in the brain of okara-fed mice compared to the control group, suggesting that this increase was caused by an increase in the expression of acetylcholine-synthesizing enzyme, ChAT. These results suggest that oral administration of okara may have the potential to be a valuable supplement in the prevention of age-related cognitive impairment associated with chronic inflammation. However, the underlying mechanisms are not fully understood. Further studies will be necessary to clarify the underlying mechanism regarding why hippocampal ChAT expression is enhanced by okara administration.

Okara contains isoflavones, which have been reported to possess the neuroprotective and anti-inflammatory effects in the investigation of cognitive impairment [36,37]. Further, genistein and the major metabolites, 7,8,4′-trihydroxyisoflavone, of daidzein have been reported to improve cognitive function by activating the cholinergic system and the ERK/CREB/BDNF signaling pathway in mice [38,39]. Lower dietary dosage such as 7.5% okara might induce the neuroprotective effects via enhanced BDNF secretion owing to isoflavone and its metabolite. On the other hand, in the case of higher dietary dosage such as 15% okara, these effects may be countered by some other properties, which could be caused by different microorganism populations in the gut.

In conclusion, the present study demonstrated that okara, a soybean by-product, may help to prevent age-related cognitive impairment in a mouse model of accelerated aging. Higher dietary doses of okara may have contributed to the alterations seen in gut microbiota. Lower dietary doses of okara improved cognitive performance and had neuroprotective effects in the hippocampus without altering gut microbiota. These findings suggest that okara has the potential to be a potent daily neuroprotective diet for the prevention of age-related cognitive impairment.

## Figures and Tables

**Figure 1 nutrients-11-02939-f001:**
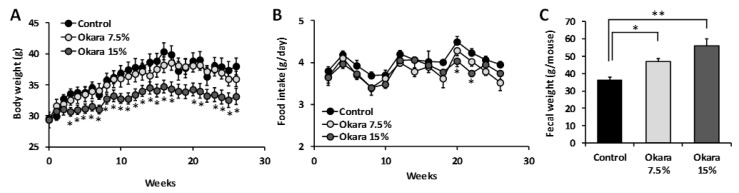
Effects of oral administration of okara on body weight, food intake, and fecal weight of senescence-accelerated mouse prone 8 (SAMP8) mice. Body weight (**A**) and feed intake (**B**) of SAMP8 mice during the feeding experiment were measured. The weight of feces (**C**) was measured at the age of 42 weeks. Data are presented as mean ± SEM; *n* = 10 mice per group; * *p* < 0.05 ** *p* < 0.01 vs. the control group.

**Figure 2 nutrients-11-02939-f002:**
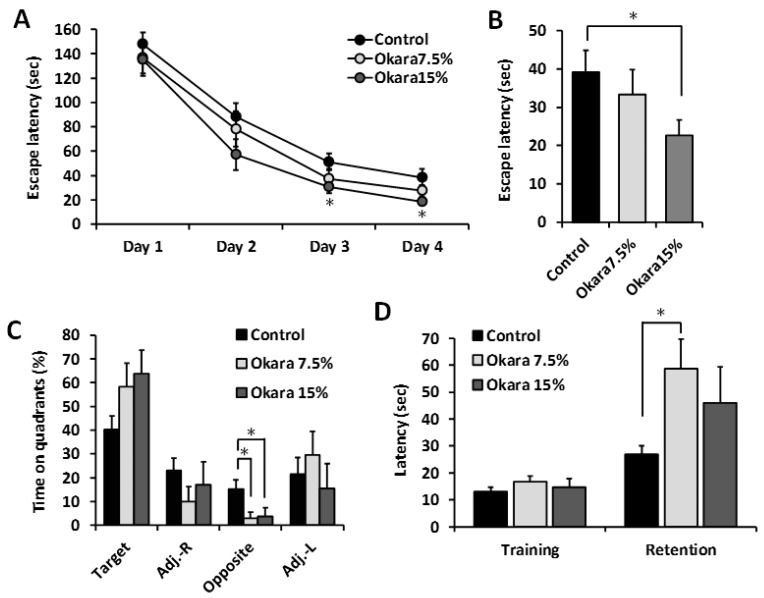
Effects of administration of okara on cognitive performance in SAMP8 mice. Escape latency during the training trials (**A**) and probe test (**B**). Time spent in each quadrant during the probe test (**C**). Step-through latency in the passive-avoidance test (**D**). Data are presented as mean ± SEM; *n* = 10 mice per group; * *p* < 0.05 vs. the control group.

**Figure 3 nutrients-11-02939-f003:**
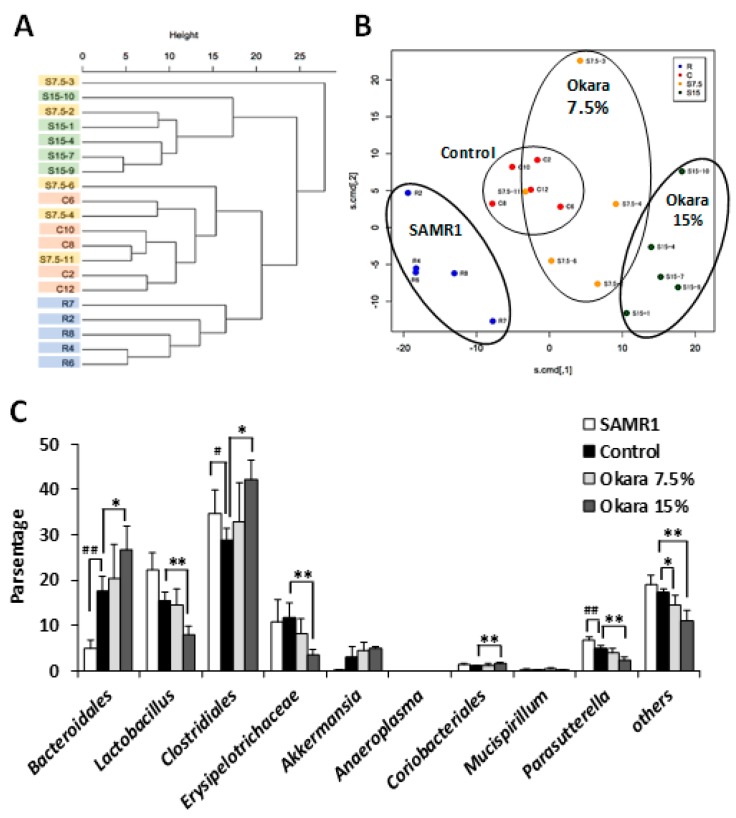
The composition of cecal microbiota in senescence-accelerated resistant mouse 1 (SAMR1) and SAMP8 control mice and the SAMP8 mice fed with a diet containing okara. (**A**) Dendrogram of correlation hierarchal cluster analysis in cecal microbiota. (**B**) Scatter plot of principal coordinate analysis. (**C**) The relative abundance of cecal bacterial composition at the phylum level. Data expressed as mean ± SEM. *n* = 5 per group. # *p* < 0.05, ## *p* < 0.01 compared with the SAMR1 group. * *p* < 0.05, ** *p* < 0.01 compared with the control group.

**Figure 4 nutrients-11-02939-f004:**
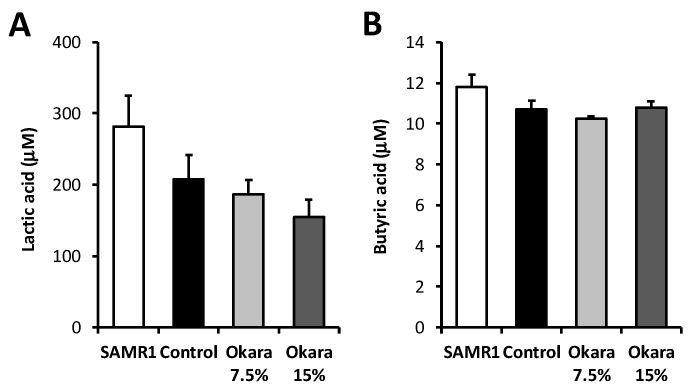
The content of lactic acid (**A**) and butyric acid (**B**) in the serum of mice. Data expressed as mean ± SEM. *n* = 5 per group.

**Figure 5 nutrients-11-02939-f005:**
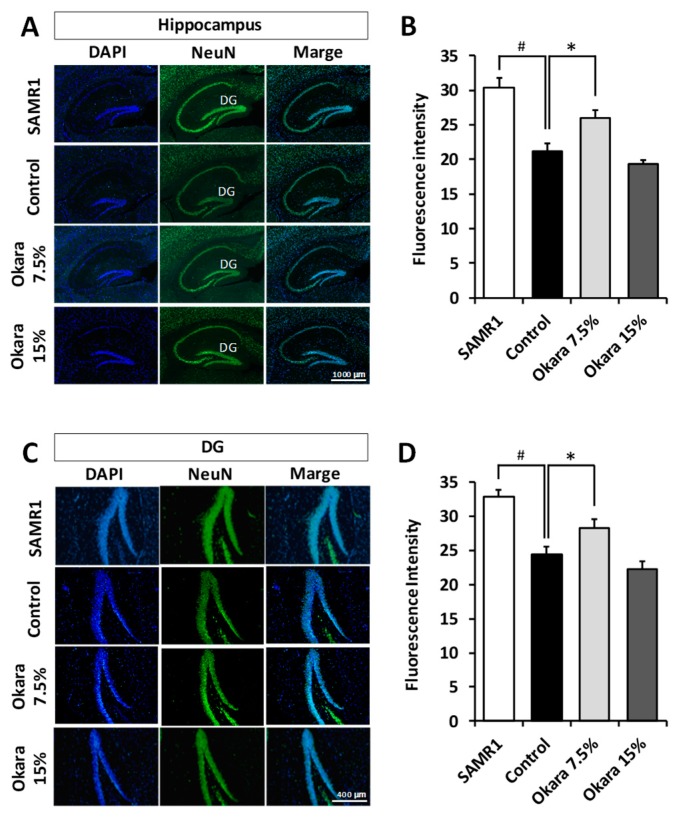
Effects of okara administration on NeuN intensity in the hippocampus and dentate gyrus (DG) of mice. Representative images of NeuN immunopositive cells (green) in the hippocampus (**A**) and DG (**C**). DAPI stained nuclei (blue). Scale bar: 100 µm. Quantification of NeuN intensity in the hippocampus (**B**) and DG (**D**). Values are expressed as the mean ± SEM (*n* = 10). # *p* < 0.05 compared with the SAMR1 group. * *p* < 0.05 compared with the control group.

**Figure 6 nutrients-11-02939-f006:**
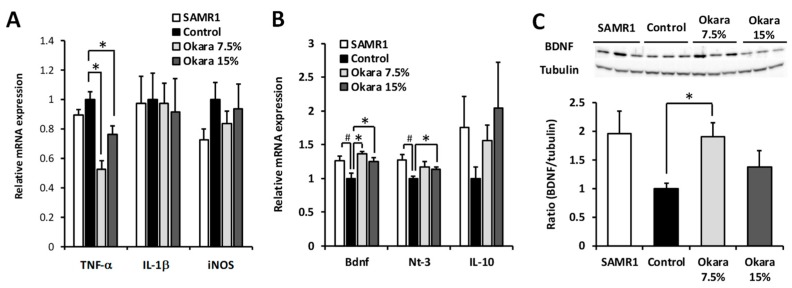
Effects of okara administration on the gene and protein expression of neuroprotective factors and neuroinflammatory factors in mouse hippocampus. Gene expression levels of TNF-α, IL-1β, and iNOS (**A**) and brain-derived neurotrophic factor (BDNF), NT-3, and IL-10 (**B**) and western blot analysis of BDNF (**C**) in the hippocampus. Data are expressed as mean ± SEM (*n* = 6), # *p* < 0.05 compared with the SAMR1 group. * *p* < 0.05 compared with the control group.

**Figure 7 nutrients-11-02939-f007:**
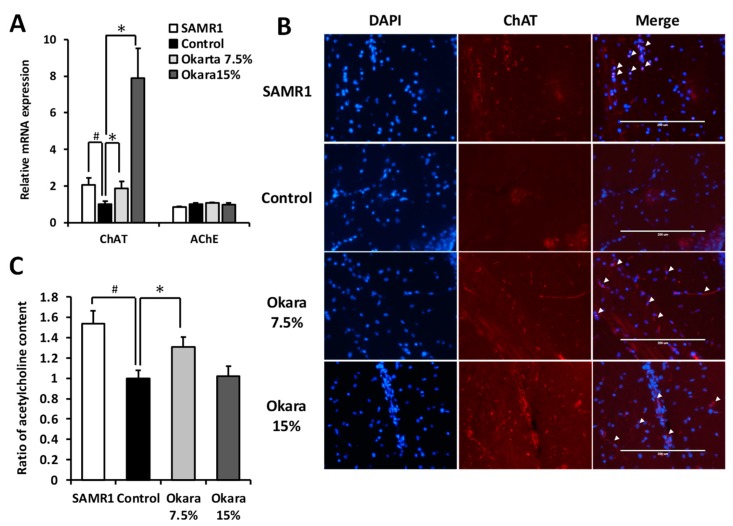
Effects of okara administration on the cholinergic systems of mice brain. (**A**) Gene expression levels of ChAT and AChE in the hippocampus. (**B**) Representative images of ChAT immunofluorescence (red) and DAPI stained nuclei (blue) in the hippocampus. Scale bar: 400 μm. (**C**) Acetylcholine content in the brain. Data presented as mean ± SEM (*n* = 10). # *p* < 0.05 compared with the SAMR1 group. * *p* < 0.05 compared with the control group.

**Table 1 nutrients-11-02939-t001:** Composition of animal diet.

Ingredients	Group		
	Control	Okara 7.5%	Okara 15%
Okara	0	7.5	15.0
Casein	14	12.2	10.4
l-cystine	0.18	0.18	0.18
Cornstarch	46.6	42.2	37.9
α-cornstarch	15.5	15.5	15.5
Sucrose	10	10	10
Soybean oil	4	2.6	1.2
Cellulose	5	5	5
Mineral mix	3.5	3.5	3.5
Vitamin mix	1	1	1
Choline bitartrate	0.25	0.25	0.25
tert-Butylhydroquinone	0.0008	0.0008	0.0008

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
