# Peer review of "Oral Administration of Okara Soybean By-Product Attenuates Cognitive Impairment in a Mouse Model of Accelerated Aging"

_nutrients, 2019, doi:10.3390/nu11122939_

Round 1
Reviewer 1 Report
None.
Author Response
We would like to thank you for the time and effort taken to review our manuscript. Best regards,
Reviewer 2 Report
The reviewers' questions and comments are properly addressed in a revised MS. Nonetheless, there are some concerns regarding grammatical errors in several spots such as lines 26, 212, and 352-360.
Author Response
We would like to thank you for the time and efforts taken to review our manuscript. We revised some grammatical errors according to your suggestion.
This manuscript is a resubmission of an earlier submission. The following is a list of the peer review reports and author responses from that submission.
Round 1
Reviewer 1 Report
Introduction:
1. This section needs to be rewritten with proper citations.
2. Introduction does not provide background in biochemical/cellular/physiological changes associated with aging.
3. Age related cognitive decline and neurodegeneration related dementia are two different and distinct processes. Introduction makes statements about these, but does not provide adequate references.
4. It must be noted that aging is a natural process and not a pathology, introduction in its current form does not provide adequate clarity. Adequate background needs to be provided regarding the role of neuroinflammation in cognitive decline.
5. Gut-brain axis is vaguely described and details with respect to how gut influences the brain is not provided. This should be done before a case is made for probiotics and prebiotics.
6. Introduction assumes that Okara is familiar to all readers, it is unclear how it is made. Statement that foods with high fiber content are prebiotic is inherently inaccurate and not factual.
7. It must also be noted that pre-biotics and probiotics are terms specifically used to describe specific substrates and specific microorganisms. Dysbiosis on the other hand is a much larger concept that relates to disruption of collective gut microbial ecology, it may or may not be related to probiotic microorganisms.
8. From the introduction it is unclear if gut dysbiosis is relevant to aging and age related cognitive decline.
Methods
1. Please describe the justification of choosing 7.5% and 15% of okara
2. Please describe the significance of using SAMP8 mice. This is a good model for early onset and accelerated pathological and non-pathological aging. It is unclear how this can model progressive chronic processes related to natural aging.
3. Were diets also normalized for the dietary fiber in Okara (1.4%). What was the final macronutrient composition of the three diets?
4. Were the SAMR1 mice also fed the three diets? Has this study been done previously? What diet were SAMR1 mice fed in this study?
5. Considering the focus on neuroinflammation, did the authors evaluate the effect of treatments on microglial populations in the brain?
Results:
1. 3.1: What was the numerical or % reduction in body weight after 3 weeks and end of the study? What was the corresponding value for fecal weight?
2. 3.1. These results suggest that administration of higher doses of okara might be effective in decreasing body weight and increasing the amount of the gut as indicated by a higher amount of feces. This is a highly speculative statement and results do not in anyway lead to this interpretation. Moreover, increased fecal mass is indicative of dietary fiber and water, not microbiota.
3. 3.2 It is unclear why data was collected only at 26 weeks? This is the case for all subsequent results and should be explained in methods/results.
4. 3.3 Unless SAMR1 mice were given the same diet/treatment, the comparison should not be made with SAMP8 mice.
5. 3.4. Again, the data should be represented in fold change with respect to SAMP8 control and SAMR1 (if relevant). Moreover, how was the fluorescence intensity calculated? Is there any data available on hippocampal atrophy?
6. 3.5. Can the authors explain non-dose dependent effect in 3.4 and 3.5? In figure 6B 15% okara resulted in a 7 fold increase in ChAT? This looks like an artifact and abnormally high. Please check statistics and mRNA calculations.
7. How was the data normalized for the qPCR? How was the loss of neuronal tissue accounted for and normalized for? Across different treatments and mouse genotypes, this must be taken into consideration for accurate analysis.
Discussion:
1. This entire section is highly speculative and overstated. It should be revised and rewritten to more accurately represent the results obtained and experiments conducted. for eg. Since no data was collected on T-reg in gut, it is presumptive to assume clostridial enrichment was responsible for this and was a determinant in lower inflammation.
2. GLUT1 expression is representative of the rate of glucose metabolism. Higher expression would indicate increased glucose transport and energy need. An increase in inflammation can dramatically increase the energy requirement (as in SAMP8 mouse) for redox homeostasis and protein quality control. It is unclear how this indicates a more efficient glucose transport? Vmax/Km of the transporter does not change, increased need upregulated GLUT1 expression allowing for more glucose uptake.
3. Line 331-335. It is highly speculative to suggest that okara modulated nfkb signaling, without any data available on p-nfkb, same is also true with TNFa.
4. Interpretation of chAT data is highly speculative.
5. This entire section will have to rewritten once all the comments are addressed.
Author Response
We appreciate the valuable comments by the reviewers and the editor. We have carefully revised the manuscript and now believe its quality to be substantially improved. Point-by-point responses to each comment can be found below.
Introduction:
This section needs to be rewritten with proper citations.Response: According to your suggestion, we rewritten the Introduction section with proper citations (Page 1, lines 37-42, Page 2, lines 65-77).
Introduction does not provide background in biochemical/cellular/physiological changes associated with aging.
Response: Thank you for your point-out. We deleted the background in in biochemical/cellular/physiological changes associated with aging and rewritten this part of Introduction section (Page 1, lines 37-42).
Age related cognitive decline and neurodegeneration related dementia are two different and distinct processes. Introduction makes statements about these, but does not provide adequate references.
Response: Thank you for your suggestion. Since this work mainly focused on the age-related cognitive decline not the neurodegeneration-related dementia, we deleted the sentences concerning the neurodegeneration-related dementia.
It must be noted that aging is a natural process and not a pathology, introduction in its current form does not provide adequate clarity. Adequate background needs to be provided regarding the role of neuroinflammation in cognitive decline.
Response: Thank you for your suggestion. We rewritten this part of Introduction section (Page 1, lines 37-42).
Gut-brain axis is vaguely described and details with respect to how gut influences the brain is not provided. This should be done before a case is made for probiotics and prebiotics.
Response: Thank you for your suggestion. We rewritten this part of Introduction section (Page 1, lines 37-42).
Introduction assumes that Okara is familiar to all readers, it is unclear how it is made. Statement that foods with high fiber content are prebiotic is inherently inaccurate and not factual.
Response: According to your suggestion, we rewritten this part of Introduction section (Page 2, lines 69-73).
It must also be noted that pre-biotics and probiotics are terms specifically used to describe specific substrates and specific microorganisms. Dysbiosis on the other hand is a much larger concept that relates to disruption of collective gut microbial ecology, it may or may not be related to probiotic microorganisms.
Response: Thank you for your point-out. We deleted the sentences regarding the word “dysbiosis” in order to avoid reader’s misunderstanding.
From the introduction it is unclear if gut dysbiosis is relevant to aging and age related cognitive decline.
Response: Thank you for your point-out. According to your suggestion, we rewritten this part of Introduction section (Page 2, lines 69-73).
Methods
Please describe the justification of choosing 7.5% and 15% of okaraResponse: Thank you for your comments. The okara content is chosen by considering the nutritional value and this value is generally reasonable.
Please describe the significance of using SAMP8 mice. This is a good model for early onset and accelerated pathological and non-pathological aging. It is unclear how this can model progressive chronic processes related to natural aging.
Response: Thank you for your comments. According to your suggestion, we rewritten this part of Introduction section (Page 2, lines 65-68).
Were diets also normalized for the dietary fiber in Okara (1.4%). What was the final macronutrient composition of the three diets?
Response: Thank you for your comments. Since we showed the wrong value of Okara composition in the data, we replaced new data. As shown in this data, the macronutrient composition is carbohydrate and dietary fiber (Page 2, lines 91-93).
Were the SAMR1 mice also fed the three diets? Has this study been done previously? What diet were SAMR1 mice fed in this study?
Response: SAMR1 were fed only normal AIN-93M diet. According to your comments, we rewritten this part of Materials and Methods section (Page 2, lines 88-91).
Considering the focus on neuroinflammation, did the authors evaluate the effect of treatments on microglial populations in the brain?
Response: Thank you for your suggestion. The data concerning the microglial populations in the brain might provide more valuable information, and we’d like to conduct this experiment in the next work.
Results:
3.1: What was the numerical or % reduction in body weight after 3 weeks and end of the study? What was the corresponding value for fecal weight?Response: Thank you for your suggestion. The body weight of SAMP8 mice reached the maximum around the 18-week-old and then decreased. Thus, it’s difficult to discuss the reduction in body weight in this work.
3.1. These results suggest that administration of higher doses of okara might be effective in decreasing body weight and increasing the amount of the gut as indicated by a higher amount of feces. This is a highly speculative statement and results do not in anyway lead to this interpretation. Moreover, increased fecal mass is indicative of dietary fiber and water, not microbiota.
Response: Thank you for your comments. According to your suggestion, we deleted the sentence, “increasing the gut microbiota as indicated by a higher amount of faces” and added the sentence, “increasing the amount of feces” (Page 5, lines 207-208).
3.2 It is unclear why data was collected only at 26 weeks? This is the case for all subsequent results and should be explained in methods/results.
Response: Thank you for your comments. Generally, the memory test such as Morris maze and Passive avoidance test cannot repeat many times because the memory of mice lives on forever in the first experiment. That’s why this experimental schedule is very reasonable.
3.3 Unless SAMR1 mice were given the same diet/treatment, the comparison should not be made with SAMP8 mice.
Response: Thank you for your comments. SAMR1 mice is generally utilized as the control mice of SAMP8 and it is a powerful data to compare the SAMP8 to SAMR1 to discuss the aging process of SAMP8.
3.4. Again, the data should be represented in fold change with respect to SAMP8 control and SAMR1 (if relevant). Moreover, how was the fluorescence intensity calculated? Is there any data available on hippocampal atrophy?
Response: Thank you for your comments. The data in fold change is also reasonable, but it’s not critical difference. In this work, we didn’t focus on hippocampal atrophy of SAMP8 mice, and future studies will be needed to clarify the effects on hippocampal atrophy.
3.5. Can the authors explain non-dose dependent effect in 3.4 and 3.5? In figure 6B 15% okara resulted in a 7 fold increase in ChAT? This looks like an artifact and abnormally high. Please check statistics and mRNA calculations.
Response: Thank you for your comments. According to your suggestion, we renewed some data in Figure 6 and Figure 7.
How was the data normalized for the qPCR? How was the loss of neuronal tissue accounted for and normalized for? Across different treatments and mouse genotypes, this must be taken into consideration for accurate analysis.
Response: Thank you for your comments. The data normalized methodology is mentioned in the sentences in the Materials and Methods (Page 4, lines 146-148).
Discussion:
This entire section is highly speculative and overstated. It should be revised and rewritten to more accurately represent the results obtained and experiments conducted. for eg. Since no data was collected on T-reg in gut, it is presumptive to assume clostridial enrichment was responsible for this and was a determinant in lower inflammation.Response: Thank you for your comments. According to your suggestion, we added new data concerning the content of lactic acid and butyric acid as Figure 4 and rewritten the Discussion section (Page 10, lines 317-328).
GLUT1 expression is representative of the rate of glucose metabolism. Higher expression would indicate increased glucose transport and energy need. An increase in inflammation can dramatically increase the energy requirement (as in SAMP8 mouse) for redox homeostasis and protein quality control. It is unclear how this indicates a more efficient glucose transport? Vmax/Km of the transporter does not change, increased need upregulated GLUT1 expression allowing for more glucose uptake.
Response: Thank you for your comments. According to your suggestion, we deleted the data regarding glucose and lactate metabolism.
Line 331-335. It is highly speculative to suggest that okara modulated nfkb signaling, without any data available on p-nfkb, same is also true with TNFa.
Response: Thank you for your comments. According to your suggestion, we deleted the data regarding nfkb western blotting data.
Interpretation of chAT data is highly speculative.
Response: Thank you for your comments. According to your suggestion, we added the data of acetylcholine content and rewritten the Discussion section (Page 10, lines 344-353).
This entire section will have to rewritten once all the comments are addressed.
Response: Thank you for your valuable comments. According to your kind suggestion, we rewritten the Discussion section by adding some new data of butyric acid and acetylcholine content.

Reviewer 2 Report
It is an interesting study that reported the protective effect of okara from age-related-cognitive decline. However, there are some concerns in the experimental design and interpretation of results.
In Table 1, the diet composition in the okara-containing group should be justified; For instance, 15% okara-containing diet has 10.4 % casein, 37.9% corn starch, 5 % cellulose while okara contains 51.1% protein, 3.6% CHO, 34.7% lipid, 1.4 % DF. The authors did not make enough discussion with regard to the effect of okara on BDNF, energy metabolism-related transporters and cholinergic enzymes and their possible role in relation to the positive effect of okara on cognition. Okara contains 66 mg/g isoflavones. Although their contents are not high, it is necessary that the cognition-improving effect of okara might be associated with isoflavones. It is not clear which component in okara has mainly contributed to cognition improvement. Although okara consists of a complex matrix, it is recommended how each component in okara contributed to anti-inflammation and cognition. The authors emphasized the potential contribution of butyrate, one of the metabolites derived from the dietary fiber in okara. I think digestion-resistant sugars such as raffinose and stachyose must have a significant role in microbiota profile and the production of short-chain fatty acids including butyrate. In fact, okara had very low DF content as mentioned in line 96.Author Response
We appreciate the valuable comments by the reviewers and the editor. We have carefully revised the manuscript and now believe its quality to be substantially improved. Point-by-point responses to each comment can be found below.
It is an interesting study that reported the protective effect of okara from age-related-cognitive decline. However, there are some concerns in the experimental design and interpretation of results.
In Table 1, the diet composition in the okara-containing group should be justified; For instance, 15% okara-containing diet has 10.4 % casein, 37.9% corn starch, 5 % cellulose while okara contains 51.1% protein, 3.6% CHO, 34.7% lipid, 1.4 % DF.
Response: Thank you for your comments. Since we showed the wrong value of Okara composition in the data, we replaced new data. As shown in this data, the macronutrient composition is carbohydrate and dietary fiber (Page 2, lines 91-93).
The authors did not make enough discussion with regard to the effect of okara on BDNF, energy metabolism-related transporters and cholinergic enzymes and their possible role in relation to the positive effect of okara on cognition. Okara contains 66 mg/g isoflavones. Although their contents are not high, it is necessary that the cognition-improving effect of okara might be associated with isoflavones. It is not clear which component in okara has mainly contributed to cognition improvement. Although okara consists of a complex matrix, it is recommended how each component in okara contributed to anti-inflammation and cognition.
Response: Thank you for your comments. According to your suggestion, we rewritten the part of this section by adding explanation concerning the active compounds in okara (Page 10, lines 324-361).
The authors emphasized the potential contribution of butyrate, one of the metabolites derived from the dietary fiber in okara. I think digestion-resistant sugars such as raffinose and stachyose must have a significant role in microbiota profile and the production of short-chain fatty acids including butyrate. In fact, okara had very low DF content as mentioned in line 96.
Response: Thank you for your comments. According to your suggestion, we added the data of butyrate as Figure 4 and rewritten the Discussion section (Page 10, lines 317-328).
